FERMILAB-PUB-21-263-T

# Many-gluon tree amplitudes on modern GPUs
# A case study for novel event generators

E. Bothmann[1*], W. Giele[2], S. Höche[2], J. Isaacson[2], M. Knobbe[1]

**1** Institut für Theoretische Physik, Göttingen, Germany
**2** Fermi National Accelerator Laboratory, Batavia, USA
* enrico.bothmann@uni-goettingen.de

## Abstract

**The compute efficiency of Monte-Carlo event generators for the Large Hadron Collider is expected to become a major bottleneck for simulations in the high-luminosity phase. Aiming at the development of a full-fledged generator for modern GPUs, we study the performance of various recursive strategies to compute multi-gluon tree-level amplitudes. We investigate the scaling of the algorithms on both CPU and GPU hardware. Finally, we provide practical recommendations as well as baseline implementations for the development of future simulation programs. The GPU implementations can be found at: https://www.gitlab.com/ebothmann/blockgen-archive.**

## 1 Introduction

The success of high-energy physics experiments at particle colliders critically depends on a detailed computer simulation of the collisions. So-called event generators link theory and experiment through an event-by-event representation of the observable final-state particles by means of Monte-Carlo integration methods. A complete event generator consists of modules, which encapsulate the physics at various energy scales [1,2]. Interactions at the highest scales, which are usually regarded as the most relevant to probing our fundamental understanding of nature, are simulated by matrix element generators. In a community effort, these programs have been nearly fully automated. They can be used for computations at tree-level [3–10], and at one-loop level [11–18] in perturbation theory and can handle a large class of fundamental and effective theories [19–21].

Such flexibility typically comes at the cost of compute efficiency. This is currently a major problem, and will be prohibitive for the high-luminosity phase of the LHC, where the projected reach in terms of final-state multiplicities and large statistics data sets poses new challenges to event generator codes [22,23]. Some event generator collaborations have therefore explored massively parallel computing techniques [24–26]. The complexity of existing generators, and the requirement to continuously provide state-of-the-art simulations for LHC physics have so far prevented the wider adoption of new computing paradigms, such as GPUs. In the past, several conceptual studies have shown substantial performance gains, both for tree-level computations [27–36] and for one-loop level computations [37,38]. In addition, the evaluation of PDFs [39] and the matrix element method [40–42] have been explored on GPUs.

Nevertheless, a production-ready GPU-enabled event generator suitable for experimental applications has not yet become available. The construction of such a generator must begin with the selection of the computational algorithm which has the best metrics in terms of flexibility and efficiency. This algorithm should also be easy to understand and implement, such that portability questions can be discussed not only by high-energy physicists, but by a somewhat larger scientific computing and computer science community.

Based on many previous perturbative QCD and Beyond Standard Model applications, we argue that such an algorithm is given by the Berends–Giele recursion [43, 44]. Due to its inherently parallel nature, the recursion can easily be thread-parallelized [6, 32]. In addition, modern computer architectures with their large memory and efficient memory access allow the trivial parallelization of the entire event loop in the Monte-Carlo integration. The computation of color factors needed for the assembly of full matrix elements can either be carried out in a factorized form, or embedded in the recursion itself [45, 46]. We refer to the factorized option as a color-ordered Berends–Giele (COBG) recursion and to the embedded one as a color-dressed Berends–Giele (CDBG) recursion. From a computational point of view, both have advantages and disadvantages, and which method is most suitable for a certain hardware depends critically on the size, bandwidth and access pattern of fast memory.

In this paper, we therefore perform a comprehensive study of the known variants of the Berends–Giele recursion and assess their strengths and weaknesses on various computing platforms. We investigate the scaling of computation time and memory usage with the number of external gluons in all-gluon amplitudes, and compare the performance of our new CPU and GPU codes to the existing general-purpose matrix element generators MadGraph [9], Amegic [5], and Comix [6]. We start from the thread-scalable Berends–Giele recursion [46], which uses the leading color approximation, and extend it to full color and helicity summed amplitudes, by using: naive sums, color sampling in the color-flow basis [47], color dressing [45] and the continuous color and helicity representation of [48]. For a realistic performance assessment of the algorithms, we do not take advantage of symmetries that are specific to all-gluon scattering, and we only take into account algorithms that generate strictly positive weights.

This paper is organized as follows. In Sec. 2 we give a brief introduction to multi-gluon amplitudes, and we review the Berends–Giele recursive relations, both in their color-ordered and color-dressed form. Section 3 describes our implementation of the various techniques in practical algorithms, bound by the constraints of existing computing platforms. The discussion of these algorithms is continued in Sec. 4, where we compare the efficiency of sampling versus summing over color and helicity, and consider computational and memory complexity. We present a comprehensive analysis of our timing studies in Sec. 5 and draw conclusions for future developments of event generators in Sec. 6.

## 2    Efficient computation of multi-gluon tree amplitudes

In this section, we recall various techniques for the efficient computation of squared $n$-gluon matrix elements. They have a rich structure and are an excellent proxy for more generic tree-level computations typically encountered in collider physics. The preferred technique for evaluating matrix elements will depend on whether it is used for analytic calculations or numerical evaluation. Furthermore, the optimal numerical evaluation technique depends on

the specific computational hardware.

## 2.1 Summation over unobserved quantum numbers

Due to QCD confinement, it is impossible to detect spin or color of individual partons. The effect of spin and color correlations at the parton-level only becomes measurable through intricate correlations in many-particle hadronic final-states. Multi-parton amplitudes summed over color and spin are therefore of highest interest for collider phenomenology. At the same time, the summation over unobserved quantum numbers is one of the major obstacles in the computation of collider physics observables. We will first discuss how these computations are performed efficiently.

We define the color and implicitly spin summed squared $n$-gluon tree-level amplitude $\mathcal{A}^{\mu_1\cdots\mu_n}_{a_1\ldots a_n}$ as

$$|\mathcal{A}(1,\ldots,n)|^2 = \sum_{a_1\ldots a_n} \mathcal{A}^{\mu_1\cdots\mu_n}_{a_1\ldots a_n}(p_1,\ldots,p_n)\left(\mathcal{A}^{\nu_1\cdots\nu_n}_{a_1\ldots a_n}(p_1,\ldots,p_n)\right)^{\dagger} \prod_{i=1}^{n} P_{\mu_i\nu_i}(p_i,k_i)\,, \qquad (1)$$

where each gluon with label $i$ is characterized by its color $a_i$ and its momentum $p_i$. The projection operator $P_{\mu\nu}$ removes the longitudinal component of the external gluons so no external ghosts particles need to be included in the calculation. In the lightcone gauge the projection operator is given by

$$P_{\mu\nu}(p,k) = -g_{\mu\nu} + \frac{p_\mu k_\nu + p_\nu k_\mu}{p\cdot k}\,, \qquad (2)$$

where $k$ is an arbitrary massless gauge vector. Motivated by the appearance of high transverse momentum production of hadrons at hadron colliders and the discovery of 3-jet production at PETRA [49–51], the first analytic expressions for 4-gluon [52,53] and 5-gluon [54,55] squared matrix elements were obtained by evaluating Eq. (1) in terms of Feynman diagrams.

To simplify the calculation and to obtain compact expressions, one can use the dyadic decomposition of the lightcone projection operator $P_{\mu\nu}$ into helicity eigenstates

$$P_{\mu\nu}(p,k) = \sum_{\lambda} \epsilon^{\lambda}_{\mu}(p,k)\epsilon^{\lambda\,\dagger}_{\nu}(p,k)\,, \qquad (3)$$

where $\lambda = \pm$ are the helicity labels of the gluon. This was first used in [56] to obtain a compact expression for the squared 5-gluon matrix elements by judicious choices of the gauge vectors $k_i$ to simplify the calculation. The squared $n$-gluon amplitude then takes the form

$$|\mathcal{A}(1,\ldots,n)|^2 = \sum_{\lambda_1\ldots\lambda_n}\sum_{a_1\ldots a_n} \mathcal{A}^{\lambda_1\ldots\lambda_n}_{a_1\ldots a_n}(p_1,\ldots,p_n)\left(\mathcal{A}^{\lambda_1\ldots\lambda_n}_{a_1\ldots a_n}(p_1,\ldots,p_n)\right)^{\dagger}\,, \qquad (4)$$

where the $\mathcal{A}^{\lambda}$ are the contraction of the original $\mathcal{A}^{\mu}$ with the helicity eigenstate $\epsilon^{\lambda}_{\mu}$.

The findings of multiple jet production at CERN (see e.g. [57,58]) motivated the calculation of 6-parton helicity amplitudes using supersymmetry relations and spinor algebra [59–61]. The obtained expressions were suitable for numerical evaluations, but making them compact required a refined treatment of color. The color decomposition introduced in [62,63] led to the definition of color-ordered amplitudes and paved the way for more refined methods to compute helicity amplitudes [64,65].

To evaluate Eq. (4), we have two choices for the treatment of the helicity and/or color sums. The traditional method is to explicitly sum over all the quantum states. The alternative technique is to replace the discrete summation by a continuous sampling through parametric integrations. For the spin states of the gluon this is familiar and involves changing helicities to polarizations. The connection between the two approaches is straightforward [48, 66]:

$$e_\mu(p, k, \phi) = e^{i\phi}\epsilon_\mu^+(p, k) + e^{-i\phi}\epsilon_\mu^-(p, k) , \tag{5}$$

so that

$$P_{\mu\nu}(p, k) = \frac{1}{2\pi} \int_0^{2\pi} \mathrm{d}\phi \, e_\mu(p, k, \phi) e_\nu^*(p, k, \phi) . \tag{6}$$

Instead of the complex polarization vector definition of Eq. (5) one can also define a real valued polarization vector (leading to real valued amplitudes for the gluon-only case). We can choose a unit vector $e_1 = \mathrm{Re}(\epsilon^+)/2$ orthogonal to the gluon momentum $p$ and a subsequent unit vector $e_2 = -\mathrm{Im}(\epsilon^+)/2$ orthogonal to $p$ and $e_1$. Then the continuous polarization vector is given by

$$e^\mu(p, \phi) = \cos(\phi) \, e_1^\mu(p) + \sin(\phi) \, e_2^\mu(p) . \tag{7}$$

In a similar fashion we can replace the discrete sum over external colors by a continuous color polarization vector which is integrated over. We first construct continuous color polarizations for the fundamental representation based on the dyadic decomposition [48, 66]

$$\delta_{ij} = \int \mathrm{d}[z] \, \eta_i([z])\eta_j([z]) . \tag{8}$$

Any real valued $\eta_i$ can be parametrized in terms of polar and azimuthal angles, such that

$$\eta_i([z]) \to \eta_i(\theta, \phi) = \begin{pmatrix} \cos\theta \\ \sin\theta\cos\phi \\ \sin\theta\sin\phi \end{pmatrix} , \tag{9}$$

and

$$\int \mathrm{d}[z] \to \frac{N_c}{4\pi} \int_0^\pi \mathrm{d}\cos\theta \int_0^{2\pi} \mathrm{d}\phi . \tag{10}$$

In this way the quark color state is represented by a spherical three dimensional unit vector.

In order to construct a dyadic decomposition for the adjoint representation, relevant for gluons, we use the identity

$$\begin{aligned} \delta^{ab} = \mathrm{Tr}(T^a T^b) &= \int \mathrm{d}[z]\mathrm{d}[\bar{z}] \, \eta_i([z])T_{ij}^a\eta_j([\bar{z}]) \, \eta_k([\bar{z}])T_{kl}^b\eta_l([z]) \\ &= \int \mathrm{d}[z]\mathrm{d}[\bar{z}] \, \eta^a([z], [\bar{z}])\eta^{b\,\dagger}([z], [\bar{z}]) , \end{aligned} \tag{11}$$

where we have defined the gluon color polarization vector

$$\eta^a([z], [\bar{z}]) = \eta_i([z]) \, T_{ij}^a \, \eta_j([\bar{z}]) . \tag{12}$$

Note that this construction differs from [48, 66], in that it uses only four instead of five integration variables and the gluon color polarization is represented by two real 3-dimensional spherical unit vectors.

Using the dyadic decompositions above, we can write the summed squared matrix element in the following form:

$$|\mathcal{A}(1,\ldots,n)|^2 = \prod_i \sumint \frac{d\phi_i}{2\pi} \, d[z]_i \, d[\bar{z}]_i \, \left| \mathcal{A}^{\phi_1\ldots\phi_n}_{[z]_1[\bar{z}]_1\ldots[z]_n[\bar{z}]_n}(p_1,\ldots,p_n) \right|^2 , \qquad (13)$$

where we have defined the color-helicity sub-amplitudes

$$\mathcal{A}^{\phi_1\ldots\phi_n}_{[z]_1[\bar{z}]_1\ldots[z]_n[\bar{z}]_n}(p_1,\ldots,p_n) = \mathcal{A}^{\mu_1\cdots\mu_n}_{a_1\ldots a_n}(p_1,\ldots,p_n) \times \prod_i e_{\mu_i}(p_i,k_i,\phi_i) \times \prod_j \eta_{a_i}([z]_i) . \quad (14)$$

They can be further simplified analytically, or evaluated numerically with the help of the Berends–Giele recursion. Furthermore, the generation of continuous (color) polarizations can easily be included in the Monte-Carlo integration over phase-space. We will investigate these questions in more detail in Secs. 4 and 5.

## 2.2 Color decomposition and color-ordered amplitudes

The decomposition of summed $n$-gluon amplitudes into helicity amplitudes $\mathcal{A}^{\lambda_1\ldots\lambda_n}$ introduced in Eq. (1) is particularly useful in analytic calculations, because it allows for a judicious choice of gauge vectors, which in turn can greatly simplify the calculation [56]. Similarly, color decompositions are useful, because they allow to factorize the color and kinematics dependent part of $n$-gluon amplitudes and calculate the color-dependent coefficients once and for all. This can be beneficial for both analytical and numerical evaluation. In this subsection, we will discuss color decompositions that are particularly useful for fast numerical computation in Monte-Carlo programs.

The most intuitive color decomposition of an $n$-gluon amplitude $\mathcal{A}_n$ is based on the adjoint representation of SU(3) [62,67,68] resulting in

$$\mathcal{A}^{\lambda_1\ldots\lambda_n}_{a_1\ldots a_n}(p_1,\ldots,p_n) = \sum_{\vec{\sigma}\in S_{n-2}} (F^{a_{\sigma_2}}\ldots F^{a_{\sigma_{n-1}}})_{a_1 a_n} \, A^{\lambda_1\ldots\lambda_n}(p_1,p_{\sigma_2},\ldots,p_{\sigma_{n-1}},p_n) , \quad (15)$$

where $F^a_{bc} = if^{abc}$. The functions $A$ are called color-ordered or partial amplitudes. If they carry a helicity label they are often simply referred to as helicity amplitudes. The multi-index $\vec{\sigma}$ runs over all permutations $S_{n-2}$ of the $(n-2)$ gluon indices $2\ldots n-1$.

Performing an explicit sum over colors in the squared amplitude, Eq. (1), leads to

$$
\begin{aligned}
|\mathcal{A}(1,\ldots,n)|^2 &= \sum_{\lambda_1\ldots\lambda_n} \sum_{a_1\ldots a_n} \mathcal{A}^{\lambda_1\ldots\lambda_n}_{a_1\ldots a_n}(p_1,\ldots,p_n)\mathcal{A}^{\lambda_1\ldots\lambda_n}_{a_1\ldots a_n}(p_1,\ldots,p_n)^\dagger \\
&= \sum_{\vec{\sigma},\vec{\sigma}'\in S_{n-2}} \mathcal{C}_{\vec{\sigma}\vec{\sigma}'} \sum_{\lambda_1\ldots\lambda_n} A^{\lambda_1\lambda_{\sigma_2}\ldots\lambda_{\sigma_{n-1}}\lambda_n}(p_1,p_{\sigma_2},\ldots,p_{\sigma_{n-1}},p_n) \\
&\qquad\qquad \times A^{\lambda_1\lambda_{\sigma'_2}\ldots\lambda_{\sigma'_{n-1}}\lambda_n}(p_1,p_{\sigma'_2},\ldots,p_{\sigma'_{n-1}},p_n)^\dagger ,
\end{aligned}
\qquad (16)
$$

where the $(n-2)!^2$ color coefficients, $\mathcal{C}_{\vec{\sigma}\vec{\sigma}'}$, are given by

$$\mathcal{C}_{\vec{\sigma}\vec{\sigma}'} = \sum_{a_1\ldots a_n} (F^{a_{\sigma_2}}\ldots F^{a_{\sigma_{n-1}}})_{a_1 a_n}(F^{a_{\sigma'_2}}\ldots F^{a_{\sigma'_{n-1}}})^*_{a_1 a_n} . \qquad (17)$$

An alternative color decomposition can be obtained by using the definition of structure constants in terms of SU(3) generators [63]

$$\mathcal{A}_{a_1 \ldots a_n}^{\lambda_1 \ldots \lambda_n}(p_1, \ldots, p_n) = \sum_{\vec{\sigma} \in S_{n-1}} \text{Tr}(T^{a_1} T^{a_{\sigma_2}} \ldots T^{a_{\sigma_n}}) \, A^{\lambda_1 \ldots \lambda_n}(p_1, p_{\sigma_2}, \ldots, p_{\sigma_n}) \,. \tag{18}$$

In this case the sum runs over all permutations $S_{n-1}$ of the $(n-1)$ gluon indices $2 \ldots n$, leading to a substantial increase in the number of partial amplitudes that contribute to the full color-helicity amplitude. Equation (16) for the explicit sum is correspondingly changed by replacing $S_{n-2} \to S_{n-1}$, $\lambda_n \to \lambda_{\sigma_n}$, $p_n \to p_{\sigma_n}$ and by making the corresponding replacements in the conjugate amplitude and the color factors, Eq. (17). Note that the minimal number of partial amplitudes needed for a complete evaluation is still $(n-2)!$ [69]. The remaining partial amplitudes can be obtained from the calculated partial amplitudes by a set of linear equations. The larger growth in the number of color coefficients and the need to construct the remaining $(n-1)$ partial amplitudes disfavors Eq. (18) for numerical computations and we will therefore only use Eq. (15).

A third color decomposition, suited especially for Monte-Carlo event generation, is the color-flow decomposition [47]. In this case, the gluon vector field is treated as an $N_c \times N_c$ matrix, $A_{ij}^\mu$, rather than a field with one color index, $A_a^\mu$. This leads to the amplitudes

$$\begin{aligned}
\mathcal{A}_{i_1 j_1 \ldots i_n j_n}^{\lambda_1 \ldots \lambda_n}(p_1, \ldots, p_n) &= \prod_{k=1}^{n} T_{i_k j_k}^{a_k} \mathcal{A}_{a_1 \ldots a_n}^{\lambda_1 \ldots \lambda_n}(p_1, \ldots, p_n) \\
&= \sum_{\vec{\sigma} \in S_{n-1}} \delta^{i_1 \bar{j}_{\sigma_2}} \delta^{i_{\sigma_2} \bar{j}_{\sigma_3}} \ldots \delta^{i_{\sigma_n} \bar{j}_1} \, A^{\lambda_1 \ldots \lambda_n}(p_1, p_{\sigma_2}, \ldots, p_{\sigma_n}) \,.
\end{aligned} \tag{19}$$

The maximum number of partial amplitudes to be evaluated for a generic color configuration, $i_1 j_1 \ldots i_n j_n$, can be as large as in Eq. (18). However, it can be shown that – when combined with color sampling – the color-flow decomposition yields the lowest average number of partial amplitudes to be evaluated per Monte-Carlo event at high parton multiplicity [47].

## 2.3 Recursive computation of (color-)helicity amplitudes

The computation of the color-ordered amplitudes can be accomplished using the Berends–Giele recursion [43], which corresponds to a dynamic programming technique that efficiently caches sets of Feynman diagrams with at least one common propagator. This propagator is dubbed the off-shell current and is computed recursively as

$$\begin{aligned}
J_\mu(1, 2, \ldots, n) = \frac{-i g_{\mu\nu}}{p_{1,n}^2} \Bigg\{ &\sum_{k=1}^{n-1} V_3^{\nu\kappa\lambda}(p_{1,k}, p_{k+1,n}) J_\kappa(1, \ldots, k) J_\lambda(k+1, \ldots, n) \\
&+ \sum_{j=1}^{n-2} \sum_{k=j+1}^{n-1} V_4^{\nu\rho\kappa\lambda} J_\rho(1, \ldots, j) J_\kappa(j+1, \ldots, k) J_\lambda(k+1, \ldots, n) \Bigg\} \,.
\end{aligned} \tag{20}$$

Here $p_i$ denote the momenta of the gluons, $p_{i,j} = p_i + \ldots + p_j$ and $V_3^{\nu\kappa\lambda}$ and $V_4^{\nu\rho\kappa\lambda}$ are the color-ordered three- and four-gluon vertices defined by

$$\begin{aligned}
V_3^{\nu\kappa\lambda}(p, q) &= i \frac{g_s}{\sqrt{2}} \big( g^{\kappa\lambda}(p-q)^\nu + g^{\lambda\nu}(2q+p)^\kappa - g^{\nu\kappa}(2p+q)^\lambda \big) \,, \\
V_4^{\nu\rho\kappa\lambda} &= i \frac{g_s^2}{2} \big( 2g^{\nu\kappa} g^{\rho\lambda} - g^{\nu\rho} g^{\kappa\lambda} - g^{\nu\lambda} g^{\rho\kappa} \big) \,.
\end{aligned} \tag{21}$$

The external particle currents, $J_\mu(i)$, are given by the helicity or polarization vectors of Sec. 2.1. A complete color-ordered $n$-gluon amplitude $A(1,\ldots,n)$ is obtained by putting the helicity/polarization dependent $(n-1)$-particle off-shell current $J_\mu(1,\ldots,n-1)$ on-shell and contracting it with the external polarization $J_\mu(n)$:

$$A(1,\ldots,n) = J_\mu(n)\, p_{1,n}^2\, J^\mu(1,\ldots,n-1)\,. \tag{22}$$

The algorithmic complexity scales as $\mathcal{O}(n^4)$ and is dictated by the four-gluon vertex. This can be further improved by decomposing the four-gluon vertex and introducing an auxiliary antisymmetric tensor field with the "propagator" [48]

$$-iD_{\mu\nu}^{\kappa\lambda} = -i\big(g_\mu^\kappa g_\nu^\lambda - g_\mu^\lambda g_\nu^\kappa\big)\,. \tag{23}$$

Then the recursive relations for the gluon and tensor fields read

$$J_\mu(1,2,\ldots,n) = \frac{-ig_{\mu\nu}}{p_{1,n}^2} \sum_{k=1}^{n-1} \bigg\{ V_3^{\nu\kappa\lambda}(p_{1,k},p_{k+1,n}) J_\kappa(1,\ldots,k) J_\lambda(k+1,\ldots,n)$$

$$+ V_T^{\nu\kappa\alpha\beta} J_\kappa(1,\ldots,k) J_{\alpha\beta}(k+1,\ldots,n) + V_T^{\lambda\nu\alpha\beta} J_{\alpha\beta}(1,\ldots,k) J_\lambda(k+1,\ldots,n) \bigg\}\,, \tag{24}$$

and

$$J^{\alpha\beta}(1,2,\ldots,n) = -iD_{\gamma\delta}^{\alpha\beta} \sum_{k=1}^{n-1} V_T^{\gamma\delta\kappa\lambda} J_\kappa(1,\ldots,k) J_\lambda(k+1,\ldots,n)\,, \tag{25}$$

where the tensor-gluon interaction is given by

$$V_T^{\mu\nu\kappa\lambda} = \frac{i}{2}\frac{g_s}{\sqrt{2}}\big(g^{\mu\kappa}g^{\nu\lambda} - g^{\mu\lambda}g^{\nu\kappa}\big)\,. \tag{26}$$

This will improve the complexity scaling of the algorithmic implementation to $\mathcal{O}(n^3)$.

The above recursion can be modified to calculate amplitudes without the need for an explicit color decomposition by using color-dressed currents instead. The most convenient form for implementing the color-dressed recursion and the techniques outlined in Sec. 2.1 is given by the color-flow basis [45], which works for both the discrete color sampling strategy of [47] and the continuous sampling of [48,66]. It is based on the use of the identity $\delta^{ab} = \text{Tr}(T^aT^b)$ for the gluon propagator. By writing $\text{Tr}(T^aT^b) = T_{ij}^a T_{ji}^b$ and assigning color indices $i$ and $j$ to the intermediate gluon, one can use the generators $T_{ij}^a$ to project the structure constants associated with the elementary interaction vertices onto fundamental indices as

$$F_{bc}^a T_{ij}^a T_{kl}^b T_{mn}^c = if^{abc} T_{ij}^a T_{kl}^b T_{mn}^c = \delta_{il}\delta_{kn}\delta_{mj} - \delta_{in}\delta_{kj}\delta_{ml}\,. \tag{27}$$

Note that we have used the conventions of [64] for the normalization of the SU(3) generators. Formally, we define the color-dressed gluon and tensor pseudoparticle currents $\mathcal{J}_{\mu\,ij}$ and $\mathcal{J}_{\alpha\beta\,ij}$ as

$$\mathcal{J}_{\mu\,ij}(1,\ldots,n) = \sum_{\vec\sigma\in S_n} \delta_{ij_{\sigma_1}}\delta_{i_{\sigma_1}j_{\sigma_2}}\ldots\delta_{i_{\sigma_n}j}\, J_\mu(\sigma_1,\ldots,\sigma_n)\,,$$

$$\mathcal{J}_{\alpha\beta\,ij}(1,\ldots,n) = \sum_{\vec\sigma\in S_n} \delta_{ij_{\sigma_1}}\delta_{i_{\sigma_1}j_{\sigma_2}}\ldots\delta_{i_{\sigma_n}j}\, J_{\alpha\beta}(\sigma_1,\ldots,\sigma_n)\,. \tag{28}$$

Denoting by $\pi$ the set $(1, \ldots, n)$ of $n$ particle indices, the following recursive relations for these currents are obtained:

$$
\mathcal{J}_{\mu\,ij}(\pi) = D_{\mu\,ij}^{\nu\,hg}(\pi) \left\{ \sum_{\mathcal{P}_2(\pi)} \mathcal{V}_{\nu\,hg}^{\kappa\,kl,\,\lambda\,mn}(\pi_1, \pi_2)\, \mathcal{J}_{\kappa\,kl}(\pi_1)\mathcal{J}_{\lambda\,mn}(\pi_2) \right.
$$
$$
\left. + \sum_{\mathcal{OP}_2(\pi)} \mathcal{V}_{\nu\,hg}^{\kappa\,kl,\,\alpha\beta\,mn}\, \mathcal{J}_{\kappa\,kl}(\pi_1)\mathcal{J}_{\alpha\beta\,mn}(\pi_2) \right\} , \tag{29}
$$

$$
\mathcal{J}_{\alpha\beta\,ij}(\pi) = D_{\alpha\beta\,ij}^{\gamma\delta\,hg} \sum_{\mathcal{P}_2(\pi)} \mathcal{V}_{\gamma\delta\,hg}^{\kappa\,kl,\,\lambda\,mn}\, \mathcal{J}_{\kappa\,kl}(\pi_1)\mathcal{J}_{\lambda\,mn}(\pi_2) .
$$

Here we have defined the color-dressed gluon and tensor pseudoparticle vertices

$$
\mathcal{V}_{\nu\,hg}^{\kappa\,kl,\,\lambda\,mn}(\pi_1, \pi_2) = \delta_{lg}\delta_{kn}\delta_{hm}\, V_{3\,\nu}^{\ \ \kappa\lambda}(\pi_1, \pi_2) + \delta_{hk}\delta_{ml}\delta_{ng}\, V_{3\,\nu}^{\ \ \lambda\kappa}(\pi_2, \pi_1) \tag{30}
$$

and

$$
\mathcal{V}_{\gamma\delta\,hg}^{\kappa\,kl,\,\lambda\,mn} = \delta_{lg}\delta_{kn}\delta_{hm}\, V_{T\,\gamma\delta}^{\ \ \ \kappa\lambda} + \delta_{hk}\delta_{ml}\delta_{ng}\, V_{T\,\gamma\delta}^{\ \ \ \lambda\kappa} , \tag{31}
$$

as well as the dressed propagators

$$
D_{\mu\,ij}^{\nu\,hg} = D_\mu^\nu\, \delta_{ih}\delta_{jg} , \qquad \text{and} \qquad D_{\alpha\beta\,ij}^{\gamma\delta\,hg} = D_{\alpha\beta}^{\gamma\delta}\, \delta_{ih}\delta_{jg} . \tag{32}
$$

The first sum in Eq. (29) runs over all partitions $\mathcal{P}_2(\pi)$ of the set $\pi$ into two subsets $\pi_{1,2}$ while the second sum runs over the set of ordered partitions $\mathcal{OP}_2(\pi)$ into $\pi_{1,2}$.[1]

# 3 Practical implementation of the algorithms

In this section we discuss different implementations of the methods laid out in Sec. 2 in actual computer code. In most cases, we provide a CPU and a GPU version, which are analyzed in terms of computational complexity and memory requirements in Sec. 4 and compared in terms of practical compute performance in Sec. 5. We use Rambo [70] to generate the phase-space points for the different methods. Obtaining a purely real algorithm reduces the memory requirements by a factor of two compared to a complex valued algorithm. In memory bound algorithms, reducing the memory of each object allows the CPU and GPU to fetch more from the cache in a single read, thus improving overall performance. Therefore, it is ideal to obtain a purely real implementation.

## 3.1 Leading color computation

The first algorithm (Tess) was originally published in [32]:

---

[1]Our implementation of the sum over partitions in Eq. (29) uses the equivalence of Eq. (3.18) and Eq. (3.16) in Ref. [45] to reduce the computational complexity.

- Helicity amplitudes are computed using the color-ordered Berends-Giele recursion, Eqs. (20) and (22). To avoid the larger memory requirements Tess does not make use of the decomposed four-gluon vertex in Eq. (24). Helicities are evaluated by sampling real polarization vectors according to Eq. (7). As a consequence, *all* numbers in the algorithm are real.

- Parallelization is achieved not only by calculating many events concurrently, but also by making use of the intrinsically parallel structure of the recursion, such that up to $n-1$ threads cooperate in the calculation of a single event.

- Global memory access is minimized by storing only event weights and gluon momenta. The shared memory per event contains all the currents and momenta needed to compute the color-ordered amplitudes and scales quadratically with the number of particles.

- In contrast to the original publication [32], the implementation has been modified to use double precision numbers, as all of the following algorithms do.

The second algorithm (BlockGen-LC) is similar to Tess, but differs in the following details:

- Only event-level parallelization is used, i.e. every threads computes the matrix element for an independent phase-space point.

- All quantities except for the internal currents of partial amplitudes are stored in the global memory of the GPU.

- Helicities can be either sampled or summed.

Both of the above algorithms rely on the leading color approximation. The number of colors $N_c$ is considered to be large, such that the color sum in Eq. (16) can be evaluated analytically. It results in a simple prefactor:

$$|\mathcal{A}_n(1,\ldots,n)|^2 = N_c^{n-2}\left(N_c^2 - 1\right)\left(\sum_{\vec{\sigma}\in S_{n-1}}\left|A^{\lambda_1\ldots\lambda_n}(p_1, p_{\sigma_2},\ldots,p_{\sigma_n})\right|^2 + \mathcal{O}\left(\frac{1}{N_c^2}\right)\right). \quad (33)$$

## 3.2 Color summation and sampling with color-ordered amplitudes

The third and fourth algorithm (BlockGen-CO$_\Sigma$ and BlockGen-CO$_{\mathrm{MC}}$) use the following approach:

- Helicity amplitudes are computed by utilizing the color-ordered Berends–Giele recursion, Eqs. (20) and (22). The two BlockGen-CO algorithms do not make use of the decomposed four-gluon vertex in Eq. (24), to avoid the larger memory requirements. Helicities are summed using real polarization vectors according to Eq. (7). As a consequence, *all* numbers in the algorithms are real.

- In the *color summed* variant (BlockGen-CO$_\Sigma$), the adjoint representation decomposition from Eq. (15) is used, which leads to the amplitude assembly formula, Eq. (16). In the *color sampling* variant (BlockGen-CO$_{\mathrm{MC}}$), the color-flow decomposition of Eq. (19) is used and color indices are generated event by event using the algorithm outlined in [6].

- All quantities except for the internal currents of partial amplitudes are stored in the global memory of the GPU. For the BlockGen-CO$_\Sigma$ variant, this includes the color matrix, Eq. (17), which we compute once using Form [71] and then read in from storage to perform the matrix element evaluations.

## 3.3  Color sampling with color-dressed amplitudes

The fifth algorithm (BlockGen-CD$_{MC}$) uses the following approach:

- Helicity amplitudes are computed using the color-dressed Berends–Giele recursion, Eq. (29), and in particular making use of the continuous color-polarizations introduced in Eq. (12). Helicities are sampled using Eq. (5), which is equivalent to Eq. (7). As a consequence, *all* numbers in the algorithm are real.

- All quantities are stored in global memory.

# 4  Computational complexity and memory requirements

In this section, we discuss the characteristics of the various algorithms introduced in Sec. 3 in terms of computational complexity and memory requirements for practical implementations of Monte-Carlo programs, which can be used for parton-level event generation for collider experiments on GPUs. We will focus on four criteria in particular: i) level of parallelization, ii) summing vs. sampling colors and helicities, iii) the scaling behavior of explicit summation and iv) memory requirements. This guides us in our selection of a set of candidate algorithms that will be studied in more detail in Sec. 5. Note that unless explicitly mentioned, the timings are measured for an event, meaning that the time to generate a phase-space point and evaluating cuts is included. As can be seen in Fig. 1, this is only relevant for the lower multiplicities and becomes negligible for the higher ones.

## 4.1  Level of parallelization

The GPU implementation of the Tess algorithm is not only parallelized at the event level, but also at the lower level of the summation over currents in the recursive amplitude computation [32]. This corresponds to a parallel computation of the (outer) sums in Eq. (20), such that up to $n-1$ threads can cooperate. The low-level parallelism can be helpful in a memory bound algorithm with several threads operating on the same data, which can be stored on the shared memory of the GPU, a fast cache managed by the program itself. The downside is a more complex implementation, due to the data sharing and the combination of different levels of parallelization.

Modern HPC GPU combine shared memory with the chip-controlled L2 cache, and have a larger throughput for loading missing data from their global memory. Therefore, the speed gains from the additional level of parallelization might be diminished. We analyze this effect in Fig. 1 which shows the computational time per event comparing Tess (orange) to BlockGen-LC (blue). For high gluon multiplicities, the two implementations perform similarly. The different scaling of BlockGen-LC at low multiplicities arises from the slightly more generic handling of phase-space generation and cuts in BlockGen-LC and is removed when we adopt the same

strategy as Tess. This is shown in dashed blue. For this reason, we only consider pure event-level parallelization in all the following implementations, avoiding unnecessary complexity. It is interesting to note the large timing improvement in Fig. 1 when comparing the original results from Tess (dashed orange) to the current results (orange). This improvement is purely due to improvements in GPU hardware, and indicates the increasing importance of modern GPUs for collider physics applications.

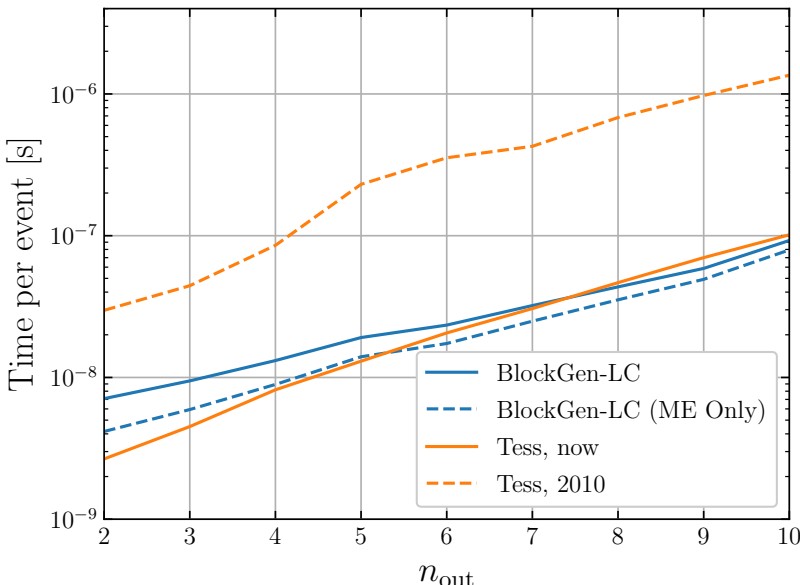

Figure 1: A comparison of the runtimes of Tess and BlockGen-LC, measured on a NVIDIA V100 (16 GB global memory, 5,120 CUDA cores, 6144 KB L2 cache). To make a more fair comparison to Tess, we include the timing for BlockGen-LC only including the time to evaluate the matrix element ("ME Only"). The results for Tess from the original publication in 2010 [32] are included as a reference to show the GPU improvements.

## 4.2 Efficiency of color and helicity sampling

Figure 2 shows the overhead for color sampling, helicity sampling, and combined color and helicity sampling as a function of gluon multiplicity in a realistic parton-level calculation. For this, we use the recursive phase-space generator of Comix. Other details of the setup are the same as in Sec. 5. We require a target precision of 1%. We measure the required number of points needed to attain this precision and plot the ratio of the number of points for the various sampling methods to the explicit sum. We observe that the relative overhead of helicity and color sampling individually is approximately constant as a function of multiplicity. However, the overhead of combined color and helicity sampling increases with increasing multiplicity. This can be understood by looking at the analytic structure of the $n$-gluon amplitudes. For 2 and 3 gluon final states, only MHV amplitudes exist. The numerator of a mostly plus amplitude is given as the fourth power of the product of the spinors for the two gluons with negative helicity. Therefore, helicity configurations can be sampled efficiently by drawing negative helicity pairs with probability proportional to the magnitude of the two-particle invariants. For higher multiplicites, the number of MHV configurations is still larger than

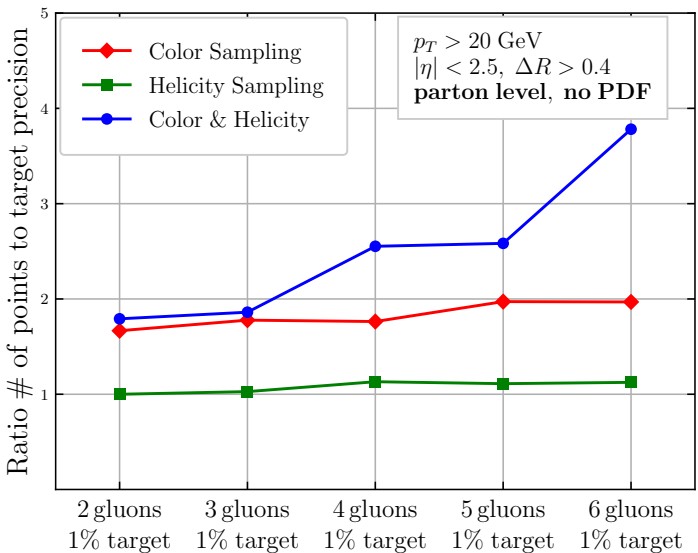

Figure 2: A comparison of the overhead for color, helicity, and combined color-helicity sampling as a function of gluon multiplicity. The ratio of the number of points needed to reach a target precision of 1% is shown relative to a calculation with an explicit sum over both colors and helicities.

the number of NMHV configurations, but the sampling according to two-particle invariants does not always approximate the true structure of the matrix element. This effect seems less pronounced in the color summed case, as can be seen in Fig. 2. However, when colors are sampled, an imperfect selection of the helicity configuration does create large fluctuations in 4- and 5-gluon final states (where MHV and NMHV configurations are present), and yet larger fluctuations in 6-gluon final states (where MHV, NMHV and NNMHV configurations exist).[2] The reason for this is that the numerator and denominator of the partial amplitudes are uncorrelated, thus an uncorrelated and imperfect sampling of color and helicity leads to reduced efficiency. Nevertheless, the timing improvements in the matrix element computation that arise from the use of sampling algorithms will never be overcompensated by a reduced convergence in the eventual Monte-Carlo integration. This point is crucial when making a choice about which algorithm to implement for GPU computing, and in particular it enables us to choose a memory-lean option.

## 4.3   Scaling of color and helicity sums

When evaluating the sum over pairs of permutations in Eq. (16) (for a given helicity configuration), the $(n-2)!$ matrix elements needed should first be calculated and stored. Then the $(n-2)!((n-2)!+1)/2$ independent summands can be calculated without re-evaluating the same matrix elements over and over. The time needed to evaluate each summand is

---

[2]We were not able to fully confirm the expected behavior in 7-gluon final states, because the corresponding color-summed prediction could not be determined within our computing budget. However, the ratio between color-helicity summed and color summed only prediction is indeed the same as in the case of the 6-gluon final state.

then dominated by loading the data corresponding to $\mathcal{C}_{\vec{\sigma}\vec{\sigma}'}$, $A^{\vec{\sigma}}$ and $A^{\vec{\sigma}'}$, and should hence be individually very small compared to the evaluation of a partial amplitude via the recursion relations in Eq. (20). However, the different scalings of the two operations, factorial for precalculating partial amplitudes vs. factorial squared for evaluating the color sum, have the consequence that there will be an $n_{\text{out}}$ for which the time required for the summation will eventually become dominant.

Figure 3 compares the time needed per event for each operation. The crossover point is found to be between $n_{\text{out}} = 6$ and 7. The inherent $\mathcal{O}((n-2)!^2)$ scaling of the color-summed algorithm means that a color-sampled algorithm will eventually become more efficient. However, since the contribution of the summing itself becomes relevant only beyond $n_{\text{out}} = 6$, this behavior might not affect practically relevant computations.

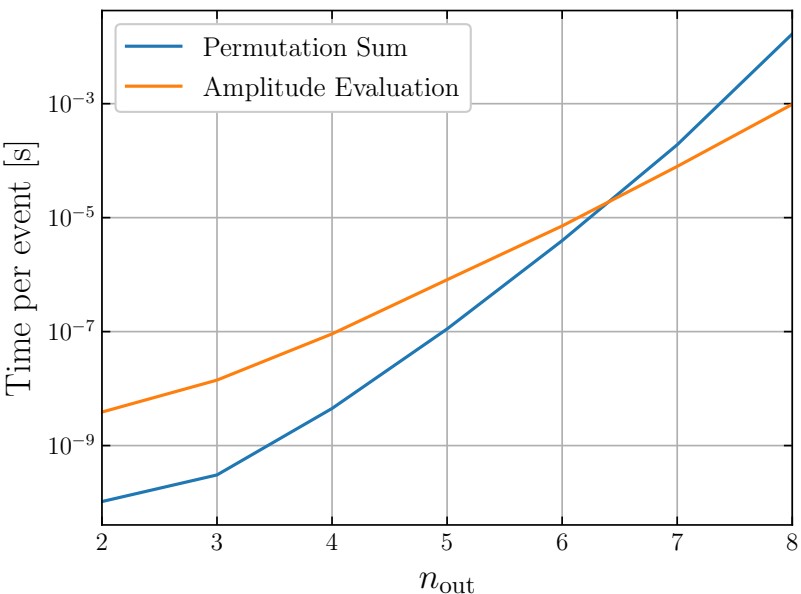

Figure 3: A comparison of the time needed to evaluate the required amplitudes versus the time needed to sum over all the permutations required as a function of multiplicity (Note that the absolute values are not directly comparable to Fig. 6, since we have used helicity sampling here.)

### 4.4 Memory requirements of summed vs. sampled algorithms

Figure 4 compares the different memory requirements on the GPU for the three different full-color algorithms BlockGen-CO$_\Sigma$, and BlockGen-CD$_{\text{MC}}$. It also includes leading color results from Tess and BlockGen-LC. The upper plot displays the global memory usage which is not event-specific, the middle plot shows the global memory usage per event, and the bottom plot shows the shared memory usage per event.

The Tess algorithm uses global memory only to store event weights and gluon momenta, which gives a small footprint in the per-event global memory plot. Everything else is stored in shared memory.

In the BlockGen-CO$_\Sigma$ algorithm, event-independent global memory is used in particular to store the color matrix, Eq. (17), leading to a rapid $\mathcal{O}((n-2)!^2)$ growth. The per-event

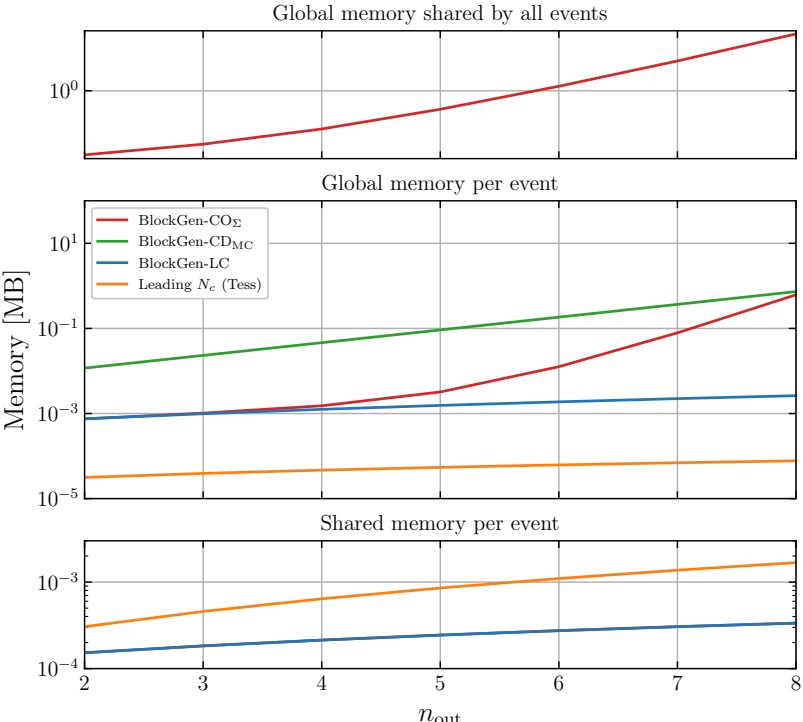

Figure 4: Comparison of memory usage in different GPU implementations. The upper plot displays the global memory required that is allocated independently of the number of events/threads. The middle plot shows the global memory per event and the bottom plot shows the shared memory per event. The shared memory per event for BlockGen-LC and BlockGen-CO$_\Sigma$ is identical.

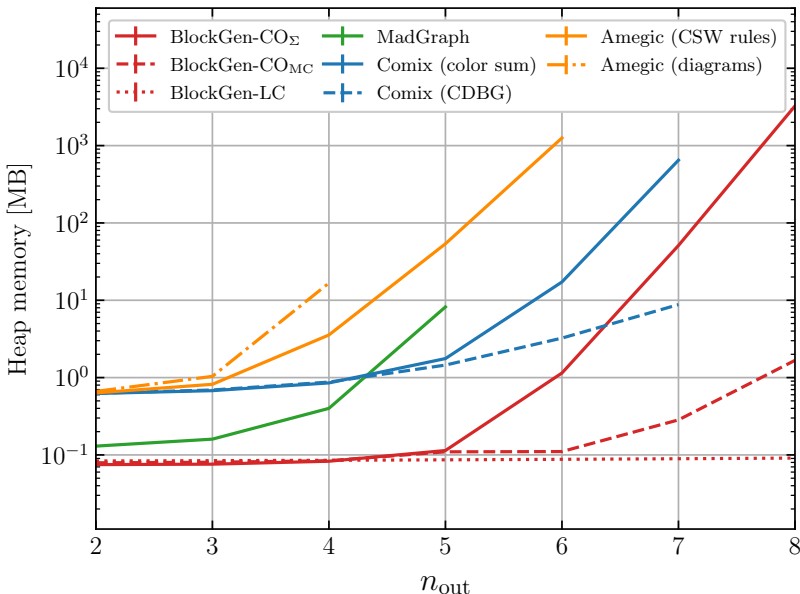

Figure 5: Single threaded CPU heap memory usage for the various algorithms (Note that the MadGraph executable size is used, since the color matrix, etc. is compiled into the executable. Therefore, the MadGraph dynamic heap memory allocations are negligible).

global memory usage is dominated by the $\mathcal{O}((n-2)!)$ growth of the number of color-ordered amplitudes $A(1, \sigma_2, \ldots, \sigma_{n-1}, n)$. Shared memory is used only for the intrinsic momenta in the recursion, such that its use grows linearly with the number of particles.

For BlockGen-CD$_{\text{MC}}$, the per-event global memory use is governed by the number of complex valued currents and tensors. Since we only need to compute the currents for ordered sets of $\pi$ in Eq. (29), the total number of objects scales as $\mathcal{O}(2^n)$. Shared memory is not used by the color-dressed approach.

We first note that the color-summed algorithm is limited to $n_{\text{out}} \leq 8$. Beyond that the memory required to store the color matrix will exceed $\mathcal{O}(10\,\text{GB})$, and hence will not fit any longer into the global memory of the GPU.[3] However, for $n_{\text{out}} \lesssim 7$ the memory use of the color-summed algorithm is below that of the color-dressed algorithm. Given that the algorithms are memory bound, we expect the color-summed algorithm to perform better than the color-dressed one for $n_{\text{out}} \lesssim 7$.

Figure 5 compares the single-thread memory usage of CPU-based implementations of our algorithms BlockGen-CO$_\Sigma$, BlockGen-CO$_{\text{MC}}$ and BlockGen-LC, with the widely used MadGraph, Comix, and Amegic codes. For MadGraph, we use version 2.9.2, while Comix and Amegic have been run as part of the Sherpa framework in its version 2.3 series [72]. For MadGraph, we use the standalone mode, supplemented with a custom Monte-Carlo loop. For Comix, we use its color summing and its color sampling mode, while for Amegic, we show the default diagram-based mode, and a mode where analytic CSW rules are employed [73], which

---

[3]At the expense of relabeling permutations we could minimize the size of the storage required for the color matrix and obtain a scaling of $\mathcal{O}((n-2)!)$, down from $\mathcal{O}((n-2)!^2)$. However, this technique would be beneficial only for large final-state multiplicities which are generally better to compute in a color-dressed approach due to the improved scaling. We therefore refrain from using it.

have been derived in [74]. We plot the RAM memory use of all codes except for MadGraph, where the code generator embeds the color matrices directly within the source code, such that the memory usage is nearly exclusively driven by the size of the compiled executable, which we therefore plot instead in this case. We find that the Comix modes and our custom algorithms have the smallest memory footprints for multiplicities of $n_{\text{out}} = 5$ and beyond, with Comix in its summed and its sampled mode using about 10 and 25 times more memory than the respective BlockGen-CO. This is likely due to Comix storing all contributing sub-currents of Eq. (28), to reduce evaluation time [6].

# 5   Performance of event generation

We now compare the algorithms described in the previous sections in terms of their practical performance.[4] Of particular interest is the time needed to generate a single event. Where appropriate, we include commonly used tools in the comparison, such as MadGraph [9], Amegic [5] and Comix [6] (the versions used are listed in Sec. 4.4).

We begin by studying single-thread performance on CPU in Sec. 5.1, proceed with the massively parallelized calculation on GPU in Sec. 5.2, and conclude with a realistic chip-to-chip comparison of (multi-threaded) CPU and GPU event generation in Sec. 5.3, in order to find the most promising hardware/algorithm combination.

For all studies, we use a partonic center-of-mass energy of $\sqrt{s} = 14\,\text{TeV}$ and require the gluon momenta to satisfy the kinematic constraints

$$p_T > 20\ \text{GeV}\,, \qquad |\eta| < 2.5\,, \qquad \Delta R > 0.4\,.$$

A parton density function is not used. For simplicity, we set the strong coupling to $\alpha_S(m_Z) = 0.118$ and use a fixed renormalization scale of $\mu_R = m_Z$.[5]

## 5.1   Comparison of single-threaded algorithms

To study the evaluation time per event of the algorithms without parallelization, we compare in Figure 6 the CPU variants of our algorithms BlockGen-CO$_\Sigma$, BlockGen-CO$_{\text{MC}}$ and BlockGen-LC, with the widely used public codes MadGraph, Comix, and Amegic ME generators, and with Tess. As described in Sec. 4.4, Comix is used in color summing and sampling mode, while Amegic is using either its default diagrammatic mode or analytic CSW rules. Figure 6 shows the evaluation time per event for these different algorithmic choices, as well as for the public codes. We find that the leading color implementations, BlockGen-LC and Tess, perform similarly. For the full-color algorithms, ratio plots show the evaluation times relative to BlockGen-CO$_\Sigma$ and BlockGen-CO$_{\text{MC}}$. Note that the implementation of BlockGen-CO$_\Sigma$ and BlockGen-CO$_{\text{MC}}$ is based on purely real numbers, cf. Sec. 3. A certain overhead compared to the public codes, which work with complex numbers, is therefore to be expected.

In both cases, Comix has the best scaling behavior due to the reuse of a maximal number of precomputed sub-currents. This reduces the naively expected factorial growth in computing

---

[4]CUDA Nsight profiling reports can be obtained from the authors upon request.

[5]Even though we only study gluon amplitudes, none of our algorithms makes use of the process-specific symmetries relating different helicity amplitudes. We do so in order to reflect the performance that can be expected from a generic, automated implementation of an event generator for arbitrary interaction model Lagrangians.

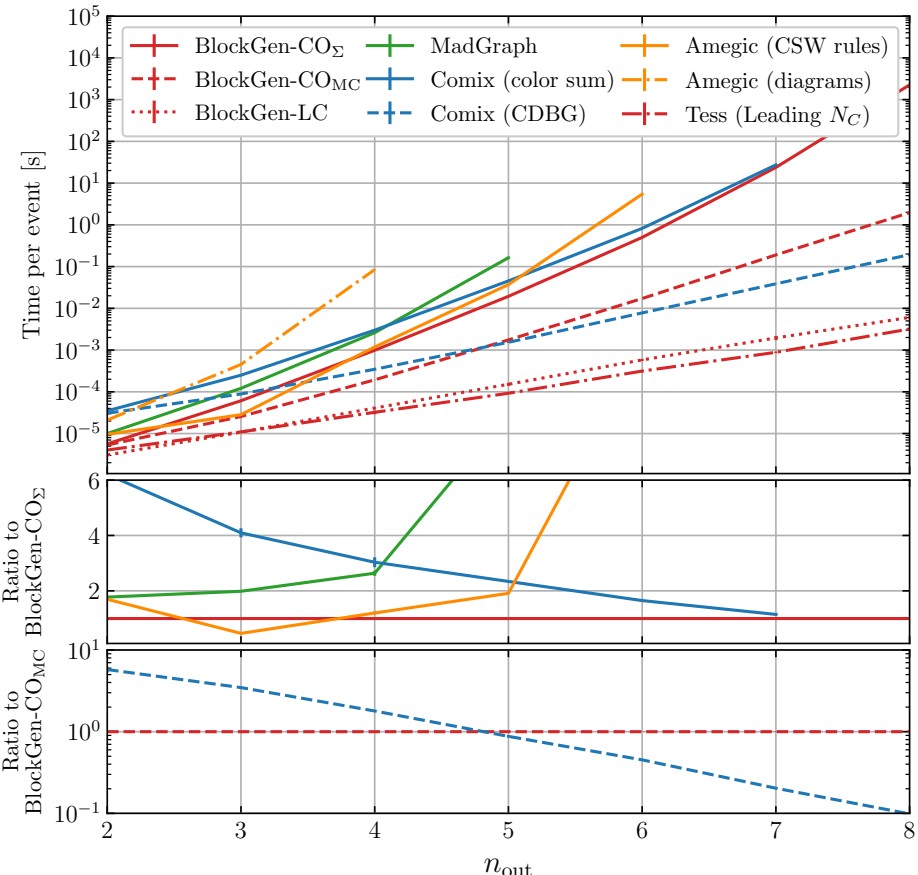

Figure 6: The timings for various CPU-based algorithms run on a single thread are compared as a function of the gluon multiplicity. The results were all generated on an Intel® Xeon® E5-2650 v2 8-core CPU (2.60 GHz, 20 MB cache).

time to an exponential growth, but comes at the cost of a larger memory footprint (see Sec. 4.4). Note that the recycling of sub-currents is not a practical option for the color-ordered algorithms. While the color-dressed algorithm only requires to store one current per *unordered set* of indices, the color-ordered algorithms (both in the color summed and color sampled variant) would require to store *ordered sets* of indices. As the number of ordered sets grows factorially with the number of members in the set, the memory required for this technique would become prohibitively large at large multiplicity. The resulting different scaling behaviors are clearly visible in Fig. 6. Hence, Comix' evaluation times are smallest for large multiplicities, falling below BlockGen-CO$_\Sigma$ at $n_{out} \approx 7$ and below BlockGen-CO$_{MC}$ at $n_{out} \approx 5$, for summing and sampling, respectively.

Taking Fig. 2 as a ballpark reference for how much a color sampled algorithm might be penalized in terms of accuracy with respect to a color summed one, we can also conclude that for single-threaded algorithms, color sampling is much faster than summing for large multiplicities.

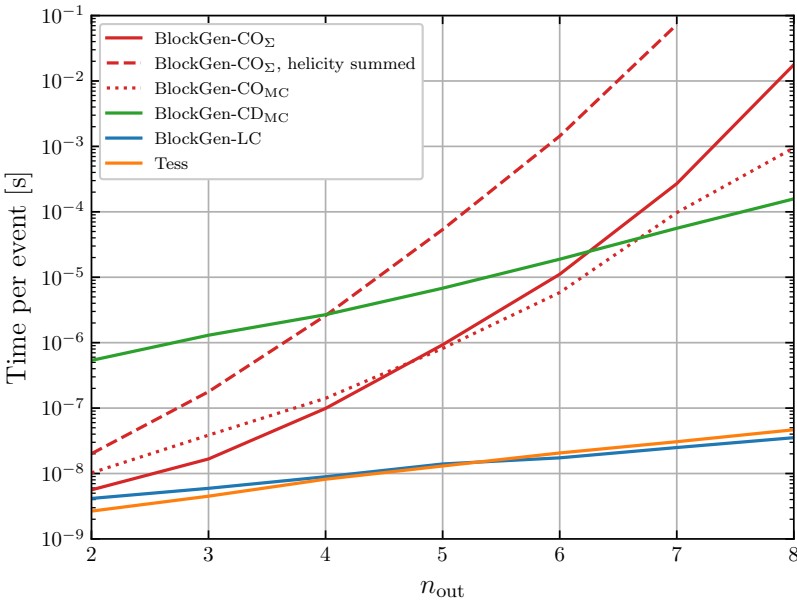

Figure 7: The timings for various GPU-based algorithms are compared as a function of gluon multiplicity. All algorithms were run on an NVIDIA V100 (16 GB global memory, 5,120 CUDA cores, 6144 KB L2 cache).

## 5.2 Comparisons of massively parallel execution on GPU

Figure 7 displays the evaluation time per event for our GPU codes as introduced in Sec. 3.

As opposed to the single thread results in Sec. 5.1, we now observe a similar performance for the color-summed BlockGen-CO$_\Sigma$ and the color-sampled BlockGen-CD$_{MC}$ algorithms. This is caused by the varying number of valid color configurations for each thread in the color-sampled case: Threads with few valid configurations need to wait for those with many. However, the scaling of the color-sampled version still is better, which is mainly due to the absence of the $\mathcal{O}((n-2)!^2)$ scaling of the color summation as discussed in Sec. 4.3.

The color-dressed BlockGen-CD$_{MC}$ exhibits an excellent scaling, but is offset by almost two orders of magnitude and only performs better than the color-ordered codes for $n_{out} \gtrsim 6$. As can be seen in Eq. (28), the color-dressed approach requires significantly more memory per current, as discussed in Sec. 4.4, such that the memory handling requires a significant amount of time during execution.

## 5.3 Comparison of single-threaded and massively parallelized algorithms

To compare the CPU-based with the massively parallelized GPU-based algorithms, we run the single-threaded Amegic (with CSW rules enabled) and Comix algorithms parallelized via MPI (Message Passing Interface) over 16 threads. The resulting performance is very close to dividing the single-thread evaluation times by 16. By making use of all the threads of our test CPU, we achieve a realistic chip-to-chip comparison to the performance of our GPU-based algorithms. However, we note that the CPU is a discontinued Intel® Xeon® E5-2650 v2 (2.60 GHz, 20 MB cache), while the GPU is a modern NVIDIA V100 (16 GB global

memory, 5,120 CUDA cores, 6144 KB L2 cache)[6]. In order to emulate a realistic chip-to-chip comparison for a similarly capable hardware, we would need to scale the number of physical CPU cores from 8 to 36, corresponding for example to an Intel® Xeon® Platinum 8360Y (2.40 GHz, 54 MB Cache). Moreover, we note again that the implementation of BlockGen-$CO_\Sigma$ and BlockGen-$CO_{MC}$ is based on purely real numbers, cf. Sec. 3. We therefore expect an overhead of about a factor of two compared to the public codes, which work with complex numbers.

The resulting times are displayed in Fig. 8. We find that for $n_{out} \leq 6$, the best-performing GPU algorithm (BlockGen-$CO_\Sigma$) has evaluation times per event which are at least an order of magnitude smaller than the best CPU evaluation times. Taking into account hardware differences, the improvement is reduced by about a factor of four, but remains substantial. For $n_{out} = 7$, the GPU code performs similarly to the CPU code. For $n_{out} \geq 7$, BlockGen-$CD_{MC}$ becomes the fastest algorithm. However, Comix achieves a similar performance on the CPU in that region, when accounting for hardware differences and the usage of complex instead of real numbers.

# 6 Conclusion

As a first step towards GPU-assisted Standard Model Monte-Carlo event generators, this study explored a variety of methods to calculate leading order $n$-gluon squared amplitudes with full color dependence. In view of increased precision requirements for high-multiplicity final-state observables at the high-luminosity LHC, we focused in particular on the scaling of the computation time with the number of external gluons. We studied different algorithms based on the Berends–Giele recursion, which differ in the treatment of helicity and color sums involved when squaring the scattering amplitude. The study was performed within a set of constraints relevant for realistic simulations. In particular, we did not take advantage of special symmetries in all-gluon scattering amplitudes and only considered algorithms that generate strictly positive weights. Our results are therefore representative of what can be expected from a full-fledged matrix-element calculator in the Standard Model and extensions thereof, which do not involve more than four-particle interactions in the Lagrangian.

Since the different algorithms vary in their scaling and memory access patterns, the algorithm with optimal performance depends on the final-state particle multiplicity. Therefore, there is no one best choice for LHC phenomenology. At low to medium multiplicity, explicit color summation combined with color ordered amplitudes, as implemented in BlockGen-$CO_\Sigma$, provides the best performance. At high multiplicity, color sampling with color dressed amplitudes, as implemented in BlockGen-$CD_{MC}$, is preferred due to the different scaling pattern. It is interesting to note that in a direct chip-to-chip comparison the existing parallel CPU-based Comix generator performs at a similar level as BlockGen-$CD_{MC}$ at high multiplicity. Given that typical event simulations at the LHC will not require more than seven jets in the final state, we conclude that BlockGen-$CO_\Sigma$ is the optimal choice to implement as a full-fledged low-to-medium multiplicity event generator on the GPU.

The next steps towards a complete Monte Carlo simulation of Standard Model events will involve the addition of quark processes and the development of a GPU-based phase-

---

[6]For additional information about the NVIDIA V100, see `https://images.nvidia.com/content/volta-architecture/pdf/volta-architecture-whitepaper.pdf`

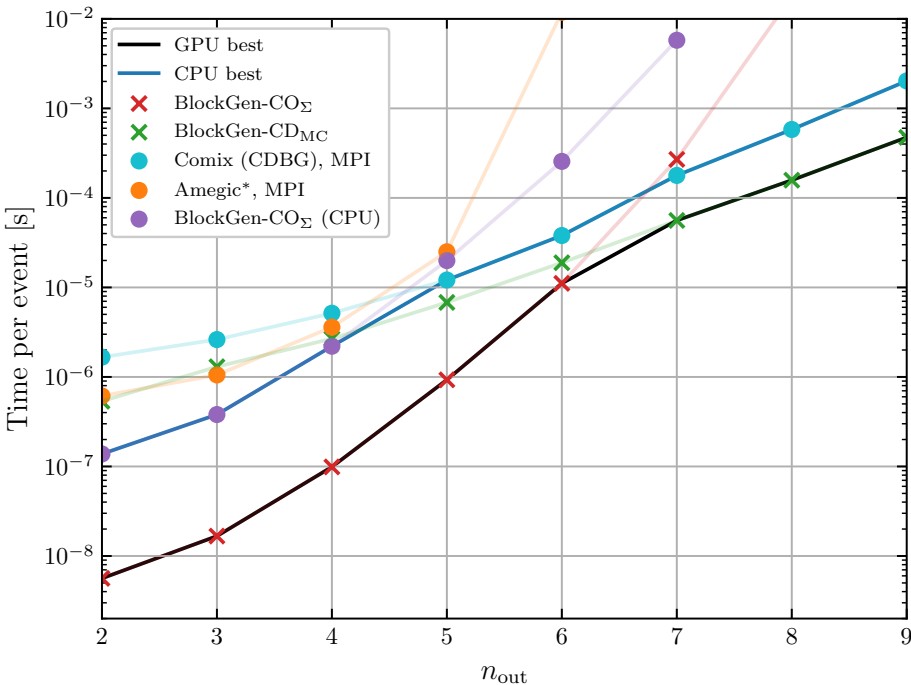

Figure 8: The timings for GPU-based (crosses) and CPU-based (dots) algorithms are compared against each other as a function of gluon multiplicity. The CPU numbers are all generated on an Intel® Xeon® E5-2650 v2 8-core CPU, (2.60 GHz, 20 MB cache), while all the GPU numbers are generated on a NVIDIA V100 (16 GB global memory, 5,120 CUDA cores, 6144 KB L2 cache). The MPI versions are run on 16 threads, and the timing for the color summed algorithm is divided by a factor of 16 to mimic the improvements that would occur from MPI. Furthermore, a modified version of Amegic is used in order to perform helicity sampling.

space generator. Since high-performance computing systems have both CPUs and GPUs associated with each node, it will be interesting to explore hybrid computation schemes with load balancing between the CPUs and GPUs available (see App. A). These hybrid schemes should be able to take advantage of the computational power available on these systems.

## Acknowledgments

This research was supported by the Fermi National Accelerator Laboratory (Fermilab), a U.S. Department of Energy, Office of Science, HEP User Facility. Fermilab is managed by Fermi Research Alliance, LLC (FRA), acting under Contract No. DE–AC02–07CH11359. MK and EB would like to thank Steffen Schumann for his support of the project, and the fruitful discussions leading towards it.

# A Proposal of a hybrid approach

In this appendix, we probe the possibility to construct a hybrid event generator code, where some parts of the computation are performed on the CPU, and other parts are performed on the GPU. Such a division of tasks can involve the phase-space generation, the calculation of phase-space weights, and the calculation of the matrix elements. In Fig. 9, we show timing ratios for these three tasks in comparison to the color-summed matrix element calculation using BlockGen-CO$_\Sigma$.

We employ the recursive phase-space generator of Comix [6] to provide the required time estimates for the phase-space tasks. This corresponds to the scenario where the CPU provides the phase-space points and weights, while the GPU only evaluates the matrix elements for those points. In this scenario, a possible bottleneck might be the communication between the CPU and the GPU. We therefore also include a time estimate for copying the phase space points to the GPU. It turns out that the memory copy is not relevant for the overall timing in this scenario. Here we assume that the current MPI implementation of Comix' phase-space generator would be changed to an OpenMP parallelized algorithm, such that the memory copy to the GPU can take place within a single process. The results in Fig. 9 show that such a division of tasks would not be viable for multiplicities $n_{\rm out} < 7$, since the GPU would mostly wait for the CPU to generate phase space points, weight them, and copy them over.

A more viable option for future development might be a scheme where the CPU only generates phase space points, while the GPU calculates both the associated weights and the matrix elements. In this scheme, the CPU would spend between about $70\,\%$ at $n_{\rm out} = 3$ and a few percent at $n_{\rm out} = 7$ of the time needed by the GPU and would thus take over a sizable share of the overall workload without forcing the GPU to be intermittently idle. Only for $n_{\rm out} = 2$, the CPU time requirement would exceed the one for the GPU, by about a factor of three. Note that these estimates assume that the phase-space weight calculation takes the same time on the GPU as it does for Comix on the CPU. If the GPU implementation leads to a sizable speed-up, the GPU would need to wait for the CPU even for $n_{\rm out} \gtrsim 2$, which might however still be an acceptable drawback to accelerate the calculation for larger multiplicities. It is beyond the scope of this work to implement a recursive GPU phase-space generator needed to further assess this scenario.

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
