# Peer review of "Many-gluon tree amplitudes on modern GPUs: A case study for novel event generators"

_SciPost Physics Codebases, doi:SciPost Phys. Codebases 3 (2022) , SciPost Phys. Codebases 3-r1.0 (2022)_

## Round 1 · Referee Report · Peter Skands (Referee 1) · 2021-9-22

Report

This is a nice and timely exposition of scaling properties of a broad set of state-of-the-art algorithms of high relevance for particle-physics applications.

Being new to SciPost CodeBases, I admit to not being sure to what extent the code repository and quality is part of my task, or is separately reviewed. I verified that the repository is available, can be downloaded, and includes a minimal README, but I could find no instructions for how to compile or test the code, so I have not included any such test or validation as part of my review. This review is therefore about the paper.

My overall conclusion is that this paper is high quality and definitely worth publishing in SciPost; nevertheless I include a number of points below which were confusing to me and/or that I think could be improved.

Requested changes

To the extent the code is considered part of this review, I would have appreciated more elaborate instructions on how to actually use what is in the repository. I could find no instructions on how to compile it, nor was there an example program illustrating a validation or use case.

On p.3, eq.(1), the colour-summed squared amplitude is defined. Given the preceding discussion and context it would be helpful to clarify that this is not only colour-summed but also spin-summed if I understand correctly, and to comment on / define the A^mu_a.

Also on p.3, it is mentioned that the findings of multiple jet production at CERN motivated the calculation of 6-parton amplitudes. I think it would be correct to include one or a few references to the relevant measurements.

In section 3 (and 4), the advantages and disadvantages of using real vs complex numbers did not become very clear to me. A real number takes up half the memory of a complex one, but does one then need the same or more numbers in total? Presumably operations with real numbers are faster than with complex ones. But again, it was not self-evident to me whether the same number of operations is needed? The use of the word “overhead” in a couple of places was ambiguous and did not make it clear which was “better” than which. According to the arguments above, I might expect a real-valued approach to take up less memory and be faster, but the converse seems to be implied a few places. That confused me.

Details are given about the CPUs used for the tests, but unless I missed it essentially nothing is said about the specs of the NVIDIA V100 used as the test bed GPU. I could look that up of course, but I think it would be useful to include reference specs for typical modern HPC GPUs, including the V100, and this could also help anchor this study better for potential future readers.

Fig.2 seemed slightly contrived to me, and given its importance for the overall conclusions I would like to request at least one complementary plot. I understand that a constant precision would not really be fair or representative of real-world applications, but there is no argument presented for why to choose these numbers, apart from “a typical SM background simulation at the LHC” without any references or even rule-of-thumb arguments provided. I'm not disagreeing, but also don't feel the argument provided is 100% convincing. The flatness of the curves appears highly dependent on the precise numerical targets chosen, and I think it will be hard for readers to use this plot to distill more general conclusions. I think two plots would be useful, such as one with constant precision (e.g., whatever can be obtained for the 6-gluon case), and then one showing the “more realistic” example already there which basically says what precision you can get if you want the curve to be flat - and explaining why that seems reasonable and encouraging in the context of realistic applications. I think that would give readers a more complete picture. The difference between the two plots would simultaneously also give the reader an idea of the scaling with the precision target, which would be useful in itself.

In the beginning of section 4.4, on p.12, figure 4 is referred to, but it does not actually appear until two pages later. I had to scroll back and forth quite a bit while reading the corresponding paragraph. Suggest to move Figure 4 to appear at the latest on the page following its first mention.

There are a couple of typographical issues with missing punctuation after BlockGen-CD_MC in the first paragraph on p.12 and after BlockGen-CO_MC on p.15.

On p.13, where analytic CSW rules are mentioned, would it not be appropriate to include a reference to the original CSW paper?

In the first line of p.18, the authors mention that they only consider algorithms that generate strictly positive weights. I'm all for that, but it was surprising to find this statement not made before the conclusions. Consider making that point earlier in the paper as well (unless I missed it).

The setup in Fig 9 in the appendix was a bit confusing to me.
Naively, I would like to compare with a reference case of everything done by GPU to see if I get a speed-up with respect to that.
But does BlockGen-CO here mean just the squared amplitude, or does it mean amplitude and phase space as previously in the paper?
Generally in the paper, it was not always completely clear to me when we are just comparing squared amplitudes, and when we are comparing phase-space sampling x squared amplitudes. I believe it is mostly the latter, and that is what the authors refer to as an 'event', but it might be worth just being more explicitly clear about that in a few places, and put fig 9 in context of how the hybrid approach compares to letting the GPU do everything.

  • validity: top
  • significance: top
  • originality: high
  • clarity: high
  • formatting: excellent
  • grammar: perfect

Author:  Enrico Bothmann  on 2022-01-11  [id 2087]

(in reply to Report 1 by Peter Skands on 2021-09-22)
Category:
answer to question
reply to objection

Dear Referee,

Thank you very much for your detailed report and your valuable suggestions to improve the preprint. We believe that we have addressed all points which you have raised in our version two of the draft, which is available on arXiv as of today and which we have just now submitted to SciPost as a resubmission.

In the following we will describe our answer to the requested changes in detail. The numbering corresponds to the individual paragraphs in the report.

  1. We have now provided instructions for compiling and running the code.

  2. We have added a definition to specify the A^mu_a that appear in Eq. (1), and we have clarified that this expression is indeed spin-summed.

  3. We have added two references to multi-jet measurements by UA1/UA2.

  4. We have added a sentence to clarify why it is indeed advantageous to obtain an algorithm that only needs real numbers. It reduces memory requirements by a factor two, and since our algorithms are memory bound, this immediately translates to a factor-of-two improvement of the runtime. This is only true because for the case at hand a single real number can replace a single complex number, without losing information. Note that this is not in general true: When adding fermions, currents need to be complex (or equivalently be represented by two real numbers, without improving performance).

  5. The key specifications of the used GPU Nvidia V100 are now listed in the paper. We have also added a link to the Nvidida V100 whitepaper, for additional information.

  6. We have created the requested plot (with a constant precision target), and in the process have devised an improved helicity sampling algorithm, which is based on the analytic knowledge of the helicity amplitudes. The basic idea is that MHV amplitudes are proportional to <ij>^4, where i and j are the labels of the minus helicity gluons in a mostly plus amplitude. This numerator factorizes from the denominator structure, which is determined by the color configuration. Therefore, an optimal helicity sampling algorithm for 2->2 and 2->3 gluon amplitudes will choose the helicity assignment proportional to the two-particle invariants. This has been implemented in order to re-create Fig. 2 and described in the text. We also show that the same algorithm leads to excellent convergence in general when colors are summed, but not when colors are sampled. This is due to the fact that numerator and denominator fluctuations are uncorrelated in non-MHV amplitudes. The effect is sizeable for 2->4 and 2->5 processes, and even more pronounced in 2->6 and 2->7 configurations. However, it is identical in the 2->4 and the 2->5 case, and in the 2->6 and the 2->7 case. This is due to the fact that only for each second additional gluon, new amplitudes of non-MHV type can emerge (NMHV in the 2->4 case, and NNMHV in the case of 2->6).

  7. The figure placement is changed as suggested, to improve readability.

  8. The punctuation issues are corrected.

  9. The original CSW paper is now referenced.

  10. That is correct, the statement should be mirrored in the introduction. It is now added there, along with the clarification that only all-gluon amplitudes are studied.

  11. We can not yet compare to a full implementation on the GPU, because we would need a proper phase-space generator implementation on theGPU, which is beyond the scope of this study. For now, we have only ported RAMBO (which is now made explicit earlier in the paper). Rambo does not allow for a meaningful comparison in the context of Fig. 9, as it is very inefficient compared to the phase-space generator of COMIX. We have rewritten parts of the appendix, and it is hopefully a bit clearer now. Regarding your second point, yes, an event refers to phase-space generation plus squared matrix element evaluation. This is now made more explicit when discussing the setup and the results.

We would again like to thank you for your thorough reading of the original draft and your helpful comments. We believe that this input has helped us to improve the quality of the draft considerably.

Best regards, the authors

---

## Round 2 · Referee Report · Peter Skands · 2022-1-28

Report

I thank the authors for the elaborations and clarifications they made in response to my report on v1 of this paper. In my opinion all concerns have been addressed, and I am happy to recommend this paper to proceed to publication.

---

## Round 2 · Author Response

We resubmit a second version to address referee comments. See the list of changes.

---

## Round 2 · List of Changes

- Provide additional instructions for compiling and running the code in the linked code repository
- Add minor clarifications and improve some formulations
- Improve layout and figure placement, fix typographical/punctuation issues
- Add two missing references for multi-jet measurements at UA1 and UA2
- Clarify why it is advantageous to use a real-number only algorithm for gluon-only algorithms
- Add technical details on the GPU used for the study
- Improve results on the comparison between sampling and summing with respect to reaching a pre-defined precision goal; related to that, add a comment on the use of an improved helicity sampling algorithm
- Extend the appendix to make our point on a possible hybrid CPU/GPU ansatz clearer; in particular note that we can not compare to a GPU-only ansatz, as a full-fledged phase-space generator implementation is still missing

You are currently on this page

Resubmission 2106.06507v2 on 11 January 2022

---

## Editorial Decision

published